

# A data-driven persistence test for robust (probabilistic) quality control of measured environmental time series: constant value episodes

Najmeh Kaffashzadeh

Institute of Geophysics, University of Tehran, Tehran, Iran

*Correspondence to*: Kaffashzadeh Najmeh (n.kaffashzadeh@ut.ac.ir)

**Abstract.** Robust quality control is a prerequisite and an essential component in any data application. That is especially important for time series of environmental observations such as air quality due to their dynamic and irreversible nature. One of the common issues in these data is constant value episodes (CVEs), where a set of consecutive data values remains
constant over a given period. Although CVEs are often considered as an indicator of sensor failure or other measurement errors and removed during quality control procedures, there are situations when CVEs reflect natural environmental phenomena, and they should not be removed from the data or analysis. Assessing whether the CVEs are erroneous data or valid observations is a challenge. As there are no formal procedures established for this, their classification is based on subjective judgement and therefore uncertain and irreproducible. This paper presents a novel test procedure, i.e., constant
value test, to estimate the probability of CVEs being valid data. The theoretical foundation of this test is based on statistical characteristics and probability theory and takes into account the numerical precision of the data values. The test is a data-driven (parametric) approach, which makes it usable for time series analysis in different environmental research domains, as long as serial dependency is given and the data distribution is not too different from Gaussian. The robustness of the test was demonstrated with sensitivity studies using synthetic data with different distributions. Example applications to measured air
temperature and ozone mixing ratio data confirm the versatility of the test.

## 1 Introduction

Millions of sensors monitor the environment every day, and their data are used in many applications such as trend analysis (Fang et al., 2013; Mills et al., 2016, 2018; Chang et al., 2017; Fleming et al., 2018; Lefohn et al., 2018) and forecast (Gardner, 1999; Zhang et al., 2012; Debry et al., 2014; Zhou et al., 2019) to provide important information on global
challenges such as climate change, air quality, soil degradation, etc. The measurement process can be interpreted as sampling from a true distribution of atmospheric state variables, for example, temperature or air pollutant concentration, at a given location. Each measured value is an estimation of "truth" that has been obtained through a set of data samples (Grant and Leavenworth, 1996). A common feature of many environmental time series is the fact that the true distribution changes with time. That makes such measurements irreproducible.





Measured data can be contaminated by various errors such as systematic, random, non-representative and gross errors (Gandin, 1988; Steinacker et al., 2011). These errors can arise from poor sensor calibration, long-term sensor drift, noise, non-resolvable processes by an observational network, and mistakes during data processing, decoding, or transmission. Some of these errors arise from unpredictable natural phenomena such as floods, fire, frost, and animal activities (Campbell et al., 2013) that cannot be documented in every detail. Although many efforts are devoted to developing advanced analytical

tools and methods, these errors can have deleterious effects on the statistical analyses. For instance, outliers, i.e., values far outside of the norm for a variable or population, can increase the error variance or reduce the power of statistical tests (Osborne and Overbay, 2004). Specifically, constant value episodes (CVEs) can decrease the normality when the assumption of a normal distribution must be satisfied, for example, in linear regression. Thus, even the most sophisticated statistical model can be vulnerable against unknown and potential erroneous data. If such errors in the data are not identified by

applying quality control (QC) procedures, the information obtained from the data will be misleading, and the results from scientific data analyses can be unreliable and biased. Therefore, robust QC procedures are an essential component in the data production chain and a requirement for having a more reliable quantification of trend or other statistical analysis.

Many research initiatives and environmental monitoring programmes have thus established standards and guidelines for QC procedures. Most of them rely on visual screening of data, and therefore personal inspection, and on manual elimination of

erroneous values based on empirical knowledge and investigator experiences. Several advanced tools such as GCE (Scully-Allison et al., 2018), CoTeDe (Castelao, 2016), AutoQC (IQuOD) and comprehensive user manuals such as QARTOD (Willis et al., 2016), WMO-AWS (Zahumensky, 2004) have been developed with precise rules to overcome this subjectivity. However, their application is often limited to a few variables or specific datasets, for example, from limited geographic regions with relatively homogenous conditions. This, in turn, can be problematic if one wants to assemble global data sets of

various environmental variables. For example, in the Tropospheric Ozone Assessment Report (TOAR), a global database with ground-level ozone measurements at more than 10,000 locations around the world was built with data from more than 30 different contributors (Schultz et al., 2017). Different QC procedures at these agencies and sites led to increased uncertainty in the assessment. At this scale of data, manual inspection methods are not only error-prone but also impractical. It is therefore desirable to develop a more generic, robust and data-driven approach for the QC of environmental monitoring

time series.

The focus of this study is to develop a QC-test for CVEs as the first element for such data-driven QC. CVEs are a common feature in air quality time series and other environmental data sets. As an example, in a specific hourly ozone time series with 35 years-long, the occurrence of the CVEs with length of 2 is 20313. Therefore, about 6.7 % of the data values are CVEs, meaning that such incidents are expected to occur naturally about 16 times per 10-days in the hourly data. The CVEs

with a longer length, e.g., 3, 4, and 5, occur 6190, 2887, and 1681 times, respectively, and so the proportion of these incidents are 4.85, 2.26 and 1.31 for 10-days hourly data time series. While they can be detected through a persistence test, a qualified judgement whether such data are erroneous or not is a difficult undertaking. If CVEs are excluded from the data (Horsburg et al., 2015; Gudmundsson et al., 2018), the results of the analysis, such as model-data comparisons (Bey et al.,





2001; Horowitz et al., 2003; Dawson et al., 2008; Emmons et al., 2010; Lamarque et al., 2012 Rasmussen et al., 2012; Tilmes et al., 2012; Im et al., 2015; Schnell et al., 2015; Lyapina et al., 2016; Sofen et al., 2016), can become biased. If the model correctly captures CVEs events, excluding the CVEs will lead to type I error. On the other hand, if CVEs originating from instrument malfunctions are included in the analysis, that will raise type I and type II errors and likely unreliable results.

This study presents a new (QC) test procedure, i.e., constant value test (CVT), which estimates the probability of a CVE

being valid data. Data users can select a threshold of an acceptable probability depending on their scientific study or data analysis task. The CVT is entirely data-driven and makes only very few assumptions about the properties of the underlying values' distribution and probability density function (Gaussian). Currently, the method is valid for data with a Gaussian frequency distribution. Possible extensions of the method are discussed in the conclusions section. In principle, it is possible to use the technique of statistical simulations to examine how the CVE probabilities change for non-Gaussian distributions.

However, this is beyond the scope of this paper. Due to its generality, the test is applicable for a wide variety of environmental variables with a serial dependency (autocorrelation). The article structure is as follows: the method (CVT) is described in Sect. 2. In Sect. 3, the approach is evaluated using synthetic data for demonstration purposes. The results of two real test cases are discussed in Sect. 4. And finally, conclusions are given in Sect. 5.

## 2 Methodology

Before describing the method, we briefly summarize some issues with existing methods. In existing QC frameworks, the persistence test is typically defined based on the minimum expected variability, but this requires prior knowledge about the true statistical distribution of the measurements. For example, Zahumensky (2004) has defined that air temperature measurements shall be flagged as "doubtful or suspected value", if the measured variable varies by less than 0.1 °$K$ over 60 minutes. Such a priori assumptions may lead to false data labelling when environmental conditions are exceptionally stable

and the true data variability is reduced for some period of time. For instance, temperature variation of 0.1 °$K$ can occur in the morning when radiative forcing is small, e.g., on a foggy day in autumn. In measurements of air pollutant concentrations longer periods of zero values can be found, if the measured concentrations are below the instrument detection limit, or if chemical conversion leads to a complete removal of a species. For example, ground-level ozone concentrations at urban sites remain zero for several hours, if there is a high level of nitrogen oxide emission.

The assessment of CVEs will also have to depend on the numerical precision or resolution (*res*), which is the number of significant digits that an observation is recorded (Chapman, 2005). For example, historical measurements of ground-level ozone at the United States air quality monitoring network in the 1980s were often reported with a resolution of 10 parts per billion (ppb). Here, it is not uncommon to find episodes of several hours when all measurements are reported as the same value, and it would be implausible to remove all of them as "erroneous measurements".

The CVT takes these considerations into account and provides a data-driven approach with very few a priori assumptions. It consists of two main procedures: first, CVEs need to be found and the length of the episodes must be recorded, then in the



second step, the probability of each CVE being a period of valid data with low variability is estimated. While the first procedure can be simply implemented by taking the differences of consecutive values, a possible complication arises, if the time series contains missing data or if the data were irregularly sampled. While the software accompanying this paper has a

provision to deal with missing data, we ignore the second issue for the purpose of this paper and require that the time series has been sampled at regular intervals. The following method description focuses on the estimation of the likelihood that two or more constant values occur in reality and are thus not necessarily resulting from measurement or data processing errors.

**2.1 Statistical background**

To describe the joint process of a given time series, we assume such a stochastic process can be represented as a multivariate

Gaussian distribution (Tong, 1990; Rencher, 2005). Let $X = (x_1 \dots x_n)$ be a series of random variables, the joint probability density function of a multivariate Gaussian distribution, $\mathcal{N}(\mu, \Sigma)$, can be written as:

$$f_X(x_1, \dots, x_n) = \frac{exp\left(-\frac{1}{2}(x-\mu)^T \Sigma^{-1}(x-\mu)\right)}{\sqrt{(2\pi)^k |\Sigma|}} \tag{1}$$

here $\boldsymbol{\mu}$ is an $n \times 1$ mean vector and $\boldsymbol{\Sigma}$ is an $n \times n$ positive definite covariance matrix. In the stationary case, without loss of generality, $\boldsymbol{\mu}$ can be assumed to be a constant and $\boldsymbol{\Sigma}$ can be represented as multiplication of a finite constant variance $\sigma^2$ and

a (auto)correlation matrix $\{i = 1 \dots n; j = 1 \dots n\}$ with $\emptyset(i, j) = 1$ if $i = j$ (diagonal) and $0 \leq \emptyset(i, j) \leq 1$ if $i \neq j$ (off-diagonal) for a given time series.

Long range approximation of an environmental time series is generally unnecessary and computationally expensive (e.g., Wincek and Reinsel, 1986; Guttorp et al., 1994; Niu, 1996; Fioletov and Shepherd, 2003; Kumar and De Ridder, 2010). Here we use an assumption that environmental time series is auto-correlated and can be approximated by an Autoregressive

(AR(1)) process (Tiao et al., 1990; Weatherhead at al., 1998, 2000; Reinsel et al., 2002). The definition of an AR (1) process, the $x_i$, i.e., data value at time $i$, can be written as:

$$x_i = const + \emptyset x_{i-1} + \varepsilon_i \tag{2}$$

here $\varepsilon_i$ is a white noise, *const* is an offset. With the assumption of AR(1) process, the correlation matrix can be approximated by one parameter $\emptyset$ since $Corr(X_i, X_{i-h}) = \emptyset^{|h|}$ (the correlation between any two points are only depended on the time

interval $h$), thus the stochastic process can be governed by three parameters, i.e., $\mu$, $\sigma^2$, and $\emptyset$.

The general likelihood of an AR(1) process can be approximated using the first-order Markov property as:

$$p(x_1, \dots, x_n) = p(x_1) \prod_{k=2}^{n} p(x_k | x_{k-1}) \tag{3}$$

where $p(x_1)$ is the density of initial state, which is not critical in this study, because the focus is placed on the probability of a consecutive state that is identical to previous value, i.e., the second term; and $p(x_k | x_{k-1})$ represents the probability

distribution of $x_k$ depending only on $x_{k-1}$. The above equation is a general form without a distributional assumption. To derive the explicit form for the Gaussian case, we start from a univariate and a bivariate probability density function:

$$f(x_{k-1}) = \frac{1}{\sigma\sqrt{2\pi}} exp\left(-\frac{1}{2}\left[\frac{(x_{k-1}-\mu)^2}{\sigma^2}\right]\right) \tag{4}$$


$$f(x_{k-1}, x_k) = \frac{1}{2\pi\sigma^2\sqrt{1-\varnothing^2}} \, exp\left(-\frac{1}{2(1-\varnothing^2)}\left[\frac{(x_{k-1}-\mu)^2}{\sigma^2} + \frac{(x_k-\mu)^2}{\sigma^2} - \frac{2\varnothing(x_{k-1}-\mu)(x_k-\mu)}{\sigma^2}\right]\right) \tag{5}$$

Then the conditional probability distribution of $X_t$ given $X_{t-1} = c$ can be derived by the Bayes' theorem and written as (see

Appendix A):

$$p(x_t \mid x_{t-1} = c) \sim N(\mu + \varnothing(c - \mu), (1-\varnothing^2)\sigma^2) \tag{6}$$

where $c$ is an arbitrary constant. The implication of such a formulation is that the resulting probability is also a function of $c$:

if the statistical model parameters $(\mu, \sigma^2, \varnothing)$ are fixed, a shorter distance of $c$ from the mean $\mu$ will result in a relatively

higher probability density than those are far away.

## 2.2 Constant value episodes (CVEs) probability

The estimation of the CVT probability consists of two steps as:

Step1. Deriving a joint probability density: for a series of (dependent) events, $A_k$ with $1 \le k \le n$, the joint density of

probability can be described through a product of multiple conditional probabilities as:

$$p(A_n \cap \ldots \cap A_1) = p(A_1)\prod_{k=2}^{n} p\left(A_k \mid \cap_{j=1}^{k-1} A_j\right) = p(A_1)\prod_{k=2}^{n} p(A_k \mid A_{k-1}) \tag{7}$$

The first equality yields from the chain rule of the joint distribution (Schum, 2001), the second equality is a special case of

an AR(1) process.

Step2. Imposing a distributional assumption to the joint probability distribution: from the Eq. (6), the probability of

consecutive values in a series with Gaussian probability density can be determined by:

$$P(CVE_{t=1, c \ne 0}) = p(x_t = c \mid x_{t-1} = c) = \int_{c-res/2}^{c+res/2} \frac{1}{\sigma\sqrt{2\pi(1-\varnothing^2)}} \, exp\left(-\frac{1}{2}\left[\frac{((c-\mu)-\varnothing(c-\mu))^2}{(1-\varnothing^2)\sigma^2}\right]\right) \tag{8}$$

The integral reflects the fact that digital data are recorded with finite numerical precision. Then according to the property of

an AR(1) process, the probability of a CVE with a length of $t$ can be calculated through $P(CVE_1)$ raising to the power of $t$-$1$

as:

$$P(CVE_{t, c \ne 0}) = \left(\int_{c-res/2}^{c+res/2} \frac{1}{\sigma\sqrt{2\pi(1-\varnothing^2)}} \, exp\left(-\frac{1}{2}\left[\frac{((c-\mu)-\varnothing(c-\mu))^2}{(1-\varnothing^2)\sigma^2}\right]\right)\right)^{t-1} \tag{9}$$

Since this equation is designed for a constant event, so the marginal probability remains a constant for each CVE. To

diminish the influence of CVE on $\mu$, they were excluded first, then the $\mu$, $\sigma$ and $\varnothing$ were calculated.

For non-normal cases, the explicit parameterization of a non-independent joint distribution is difficult to derive due to

mathematical challenge and often does not have a closed form. The nonparametric alternative is to use empirical distribution

(Epanechnikov, 1969; Waterman and Whiteman, 1978) or kernel distribution (Hwang et al., 1994; Duong and Hazelton,

2005), but this approach is not desirable for database management at this stage, because it is difficult to develop a unified

framework that is adequate for all situations. Besides, the empirical distribution estimates a probability without taking into

account of auto-correlation, i.e., independent of the adjacent data points.





The AR(1) assumption can be relaxed by increasing the order of autocorrelation without too much complexity. For example, for an AR(2) process, one could specify the covariance matrix in Eq. (1) as:

$$\Sigma = \begin{vmatrix} \sigma^2 & \sigma^2\emptyset_1 & \sigma^2\emptyset_2 \\ \sigma^2\emptyset_1 & \sigma^2 & \sigma^2\emptyset_1 \\ \sigma^2\emptyset_2 & \sigma^2\emptyset_1 & \sigma^2 \end{vmatrix} \qquad (10)$$

and modify Eq. (7) in step 1 as:

$$p(A_n \cap \ldots \cap A_1) = p(A_1)\,p(A_2|A_1)\prod_{k=3}^{n} p(A_k\,|\,A_{k-1}, A_{k-2}) \qquad (11)$$

then update the conditional probability parameterized by $(\mu, \sigma^2, \emptyset_1, \emptyset_2)$ in step 2. The more general extension of the autoregressive model is out of the scope of this study and can be referred to Box et al., (2015).

For the variables with extra incidences of zero such as nitrogen oxides (NO) and ozone the lower interval of the integration
in Eq. (9) was changed from $c$–$res$ to 0. Note that in reality "zero" values in measurements may actually be recorded as small positive or negative numbers. This detail is ignored in the following, because there is no universally applicable correction available. Some datasets may require a linear or non-linear bias correction, while for other datasets a simple cutoff, e.g., set to zero if |value| < threshold, may be more appropriate.

## 3 Model sensitivity test

The $P$ in Eq. (9) is affected by the parameters $\mu$, $\sigma$, $\emptyset$, $c$, $t$, and $res$. A simulation study was developed to evaluate the sensitivity of $P$ to each parameter. Several experiments were conducted by generating a synthetic data series to demonstrate the influence of each parameter. For each experiment, the CVT was performed over a range of possible values.

A set of first-order autoregressive, AR(1), time series with hourly time steps and a length of 240 values (10 days) was generated using Eq. (2) and a random noise generator. As a reference case *(ref)*, we set $\mu = 10$, $\sigma = 4$, and $\emptyset = 0.8$. The
numerical precision was defined as 0.01. Four sets of CVEs with the same length ($t = 3$) were added to this time series. The distance of the CVE from the mean, i.e., $c$-$\mu$, was given as 0, 1, 2, and $3\sigma$ (see Fig. 1). In this figure, four CVEs are illustrated with a colour code, i.e., red, blue, cyan and black, which are shown with boxes. The $P$ varies from $7.67\times10^{-6}$ for the first CVE to $4.77\times10^{-7}$ for the fourth (last) CVE. As stated in Sect. 2.1, the value of $P$ decreases as $c$-$\mu$ increases. CVEs which are further away from the mean are less likely to occur in nature.

To assess the effect of $t$ on $P$, a set of values ranging from 2 to 10 were selected for the $t$. All other parameters were fixed as in the baseline time series. As expected from Eq. (9), the $P$ decreases exponentially with $t$ (panel (a) in Fig. B1). Note that the slope of this exponential decrease depends on $c$-$\mu$. The larger the $c$-$\mu$, the larger would be the slope. That is in agreement with Fig. 1, where the $P$ decreases as the CVEs gets further from the mean. However, the probability of finding two consecutive data points with the same value is about 1:300, i.e., in a year-long time series such incidents are expected to
occur naturally about once per year if the sampling resolution is daily and about 25 times if the sampling resolution is hourly.

To investigate the non-linear influence of $\sigma$ on $P$ in Eq. (9), a range of values, i.e., 0.1, 0.2, 0.3, 0.4, 0.5, 1, 2, 3, 4, 5, 10, 20, were set as $\sigma$, while other parameters remained unchanged. In this scenario, the $P$ changes from $1.22\times10^{-2}$ for the smallest $\sigma$





to $8.93\times10^{-8}$ for the largest one (panel (b) in Fig. B1). By using Eq. (9), it thus becomes possible to estimate likelihoods for naturally occurring CVEs for datasets with different variability, in contrast to classical approaches, which use a fixed

variability threshold.

The most interesting parameter to consider in the CVT is the lag-1 auto-correlation ($\emptyset$). A sensitivity experiment with several additional time series was performed to assess the sensitivity of $P$ with respect to $\emptyset$ (panel (c) in Fig. B1). In this figure, $P$ ranges from $1.23\times10^{-10}$ to $2.5\times10^{-3}$. The larger the $\emptyset$ (i.e., stronger persistence), the larger would be the probability of naturally occurring CVEs. The estimated probability is very sensitive to $\emptyset$ as it approaches 1. At the limit value of 1 Eq.

(9) is undefined. If $\emptyset = 0$, the time series only consists of noise, so it is less probable to get any CVEs.

Another parameter influencing $P$ is the data digital resolution (*res*) or precision, where the data have been recorded in a fixed numerical precision (number of decimals) or as integers with possible rounding to the nearest multiple of 5, 10, etc. This parameter is shown in Eq. (9), where the resulting probability is integrated over the range of values from $c - res/2$ to $c + res/2$.

To investigate the sensitivity of the $P$ to the *res* parameter, the baseline time series was resampled by using several resolutions, i.e., 0.0001, 0.0002, 0.0005, 0.001, 0.002, 0.005, 0.01, 0.02, 0.05, 0.1, 0.2, 0.5, 1, 2, and 5. As shown in panel (a) in Fig. B2 for the example of *res* = 5, larger *res* leads to additional CVEs and it becomes harder to distinguish valid episodes from erroneous incidents. But here, to isolate influence of *res* on $P$, first the data were truncated to a new resolution, then the CVEs were added to the data. The CVT results are shown in panel (b) Fig. B2, in which the $P$ changes from $4.77\times10^{-11}$ to

$7.57\times10^{-1}$. That shows by increasing the *res*, the $P$ increase meaning that if the data are recorded in a coarse resolution, there is a higher chance to count those data as a valid data.

An experiment with several scaling factors, i.e., *fc* = 0.1, 0.2, 0.5, 1, 2, 5, and 10, were performed to check the robustness of the CVT to the different data transformations. In this experiment, the CVEs were added first, then the scaling, i.e., *x(t)×fc*, was applied, and the data were truncated to a new numerical resolution given by *res×fc*. Scaling changes other parameters

such as $\mu$ or $\sigma$, except $\emptyset$ which remains invariant. Panel (d) in Fig. B1 shows the robustness of the CVT output ($P$) with scaling. It is important to note that Eq. (9) is robust to the other data transformation such as normalisation and standardization (see Appendix C).

A combined sensitivity analysis was performed to illustrate the effect of the parameters $\sigma$, $\emptyset$, and *res* in Eq. (9), i.e., the conditional probability for two consecutive values, was evaluated over a range of conditions ($\sigma$ and $\emptyset$ from 0.01 to 0.99, and

*res* of 0.01, 0.1, and 0.5) with $\mu\text{-}c = 0$. The results are shown in Fig. 2 and can be interpreted as an upper limit for $P$ that two successive values are valid data, because $\mu\text{-}c = 0$ represents the maximum of the Gaussian distribution in Eq. (9). Using the chain rule from Eq. (11), these results can easily be extrapolated to longer CVEs. As Fig. 2 shows, the probability of finding two valid consecutive data points with the same value decreases rather quickly with increasing standard deviation $\sigma$. The $\emptyset$ has limited influence up to values around 0.7. Above this threshold, the likelihood of a two-value CVE increases drastically.

A coarser numerical resolution makes it more likely to encounter constant values in reality. At *res* similar to $\sigma$, the length $t$ of the CVE will have to be much larger than 2 to reliably classify it as erroneous. In practical applications, one would generally





set a threshold for the acceptable probability first. The information provided in Fig. 2 can then help to identify typical parameters of the time series, where this threshold will be reached.

## 4 Results and discussion

Two data time series were retrieved from the Tropospheric Ozone Assessment Report (TOAR) database (Schultz et al., 2017) to illustrate the practical use of the CVT. This database holds in-situ measured data time series for ground-level ozone in hourly time resolution. We selected the time series of ozone mixing ratio at the Azusa station (34°8′ N, 117°55′ W) in California that has data from the 1980, when the data were recorded with a resolution of 8 ppb, depending on the environmental conditions. Besides, the TOAR database contains data for meteorological variables at some stations. We

selected one temperature time series at the Cape Grim station, Tasmania (40°68′ S, 144°69′ E). This station is located at the altitude of 94 m directly on the coast, and it is a Southern Hemisphere background site with an extensive record back into the 1980. The station primarily measures air which has passed over the Southern Ocean for several days. So, temperature variations at this site are often of small amplitude.

### 4.1 Temperature

Temperature is one of the key variables relevant to air quality research. For example, temperature is often used as a primary predictor for smog-related air quality. For demonstration of the CVT in a real data situation, 10-days of a temperature time series was selected. The $\mu$, $\sigma$ and $\emptyset$ of the selected 10-days time series are 12.55, 1.59, and 0.94, respectively. The recorded numerical resolution of the data is 0.01. The time series along with the probability ($P$) of each value being a valid observation is shown in Fig. 3. Altogether, 18 CVEs are visible in Fig. 3; 15 of them with $t = 2$, 2 with $t = 3$, and 1 with $t = 4$.

The CVEs occur at more or less regular times in the early morning, e.g., 04, 05 and night-time hours, e.g., 10, 21, 22 and 23 (see Fig. 4). That can be because of the local meteorological phenomena at this site where the temperature has little variance. Therefore, these CVEs are less likely to be erroneous data.

The probabilities estimated by the CVT are above 0.2 in most cases, which means that, when the CVEs were to be flagged as erroneous data, one would err in one out of five cases and throw out valid measurements. The CVE on January, 18[th], yields

the lowest probability (0.008), in line with the expectation of the human data analyst, because it is a sparse CVE with four consecutive values ($t = 4$). This example illustrates that it will generally be impossible to define a universal threshold for $P$, but that this instead depends on the use case. For example, in a data quality control workflow at the originating institution, one may decide to rule out data with $P < 10^{-4}$, but have a data curator cross-check the measurements with larger $P$. In contrast, when these data are integrated in a larger analysis consisting of many stations, one might apply the CVT to rule out

data with $P < 10^{-3}$ or even $P < 10^{-4}$ to increase the statistical robustness of the analysis.



## 4.2 Ozone

Ozone near the ground is an air pollutant that is detrimental to human health and vegetation growth. Ozone measurement techniques have evolved over time, and it can therefore be challenging to assess the data quality of a decade-long monitoring data set, such as that from the Azusa station in California, U.S. (34°8′ N, 117°55′ W) that contains a relatively long data record from 1980 to 2016.

Figure 5 shows a 10-days examples from this measurement series for the year 1990 with the $\mu$, $\sigma$, and $\emptyset$ of 16.55, 17.32, and 0.79, respectively. During the early period, the data were reported in a low resolution, here an interval of 8 ppb. As a consequence, the time series contains many CVEs and most of them are probably valid. In contrast, for the year 2012 when the data are recorded in a higher data resolution, i.e., 1 ppb, the number of the CVE is small (see Fig. D1). As mentioned in the introduction, urban ozone time series often show very low values (effectively zero), which are however recorded as small positive or negative values, here +2 ppb. Figure 5 shows the probabilities between $3.12\times10^{-10}$ and 1 for these episodes, which have values of 2 ppb. There are also three CVEs, with large $t$ ($\geq 8$) and very low ozone mixing ratios of 2 ppb, which are shown with red circles in Fig. 5. This illustrates the issue of zero-bounded data mentioned in the methodology. The CVT can recognize such cases and the associated probabilities are $3.12\times10^{-10}$, $2.22\times10^{-7}$, and $2.48\times10^{-8}$, for the CVE1, CVE2 and CVE3, respectively. That would prevent such (valid) values from being flagged or filtered as an erroneous data. In contrast, the time series in Fig. 6 (for the year 2011) exhibits sparse occurrence of episodes, i.e., 21 CVEs where 17, 2, 1, and 1 CVEs with the t = 2, 4, 7 and 9, respectively. In most cases (17 episodes), the CVEs consist of only two consecutive values ($t = 2$). The estimated probability for these cases is between $2.15\times10^{-2}$ and $9.9\times10^{-2}$ (Fig. 6). One episode during 18[th] Nov 2011 consists of nine constant values of 2 ppb. The estimated $P$ for that incident is $4.6\times10^{-1}$, and this episode would indeed raise suspicion of a trained data analyst because such a pattern in the data would require a rather special explanation (see Fig. D3). Figure 5 also illustrates the problem with missing data values that was mentioned in the beginning of Sect. 2. On 18[th] Nov, there is a portion of gap in the time series, where the data point has been excluded, and the values to the left and right of this episode are identical. If these values were not treated correctly, those will be counted as a CVE episode with a length of eight and probability of $2.58\times10^{-7}$, which is shown with an orange circle in Fig. 5. Although such incidents could raise suspicions, they are not (and should not be) detected by the CVT. An independent test needs to be designed for such situations.

## 5 Conclusions

Environmental time series is a valuable and essential data source for scientific assessment of air quality and climate change. One of the issues in these data is the occurrence of the constant value episodes (CVEs). These episodes are often considered as an indicative of sensors' malfunctions or other measurement errors, and excluded from the data via quality control (QC) procedures. However, these episodes can be due to the natural environmental phenomena and they are indeed valid observations. Thus, distinguishing whether the CVEs are erroneous or valid data accompanied by a large uncertainty.



This study presented a theoretical concept and evaluation for a data-driven constant value test (CVT), which takes into account the typical evolution of environmental state variables such as air temperature or ozone mixing ratio as time series with serial dependence. Based on the calculus of a marginal, joint and conditional Gaussian probability density, one can estimate the probability of constant value episodes (CVEs) of length $t$ to occur in reality and use this information to flag data as potentially erroneous. The threshold for such flagging needs to be selected by the data analyst. Together with the batch size for processing pieces of the time series (in our examples, the full length of the depicted data was used; for practical applications on longer time series, we recommend sample sizes on the order of 100), these are the only a priori parameters needed. Examples with synthetic and real data demonstrate that the CVT captures many aspects, which a trained data analyst would consider in the QC of such time series. But as a data-driven approach, it will reveal data inconsistencies (here, CVEs due to measurement or data processing errors) in automated data processing workflows, and it may assist manual data quality control by making it possible to provide a fine-grained warning to the data analyst that something may be wrong with the measurements based on a probabilistic score.

The test first detects CVEs by testing for zero difference. Then, it evaluates the distribution parameters mean ($\mu$), standard deviation ($\sigma$), and lag-1 auto-correlation ($\emptyset$), and the numerical resolution of the data in user-defined portions (batches) of the time series. Given these parameters, the conditional probability for two consecutive identical values is computed and integrated over the interval given by the numerical resolution of the recorded data. Using the chain rule for the non-independent conditional probability, this probability can easily be scaled to arbitrary lengths of CVEs.

The novelty of this approach is its foundation in statistical theory and the concept of estimating a probability of a data sample to occur naturally. This distinguishes the method from classical approaches where more or less arbitrary thresholds need to be defined prior to testing. Such pre-defined thresholds can be dangerous if conditions change, for example when the same thresholds are applied to data from different world regions, climatic zones, or seasons. The method is robust against such changes and its application requires little background knowledge about the specific dataset under investigation. The method is therefore well-suited for having robust and automated QC systems, for example in smart sensor networks, where human intervention is not feasible.

**Code availability.** The Python 3.7 code of the methodology will made be available to reader under Creative Common license on the GitHub repository of the author.

**Data availability.** Schröder et al. TOAR Data Infrastructure; https://doi.org/10.34730/4d9a287dec0b42f1aa6d244de8f19eb3

**Competing interests:** The author declares no competing interests.

**Disclaimer:** Publisher's note: Copernicus Publications remains neutral with regard to jurisdictional claims in published maps and institutional affiliations.

**Financial support.** This research had been supported by ERC-2017-ADG 787576.



**Acknowledgements**

The scientific and technical support, various comments and suggestions by PD. Dr. Martin G. Schultz have greatly improved
this paper. The Australian Bureau of Meteorology for providing the temperature time series data from Cape Grim and the
U.S. EPA for providing the ozone time series at Azusa are appreciated.

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

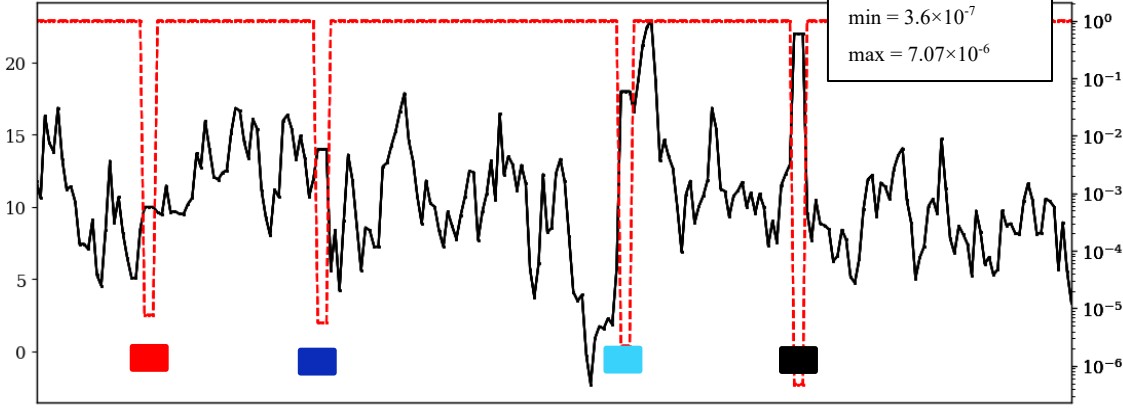

**Figure 1. A synthetic AR(1) time series with Gaussian data distribution and four arbitrarily selected CVEs of length $t = 3$ with $\mu = 10$, $\sigma = 4$, $\emptyset = 0.8$, and c-$\mu = 0$, 4, 8, and 12, respectively. The CVEs are shown using a colour code, i.e., red, blue, cyan and black. The numerical precision (*res*) is chosen as 0.01.**






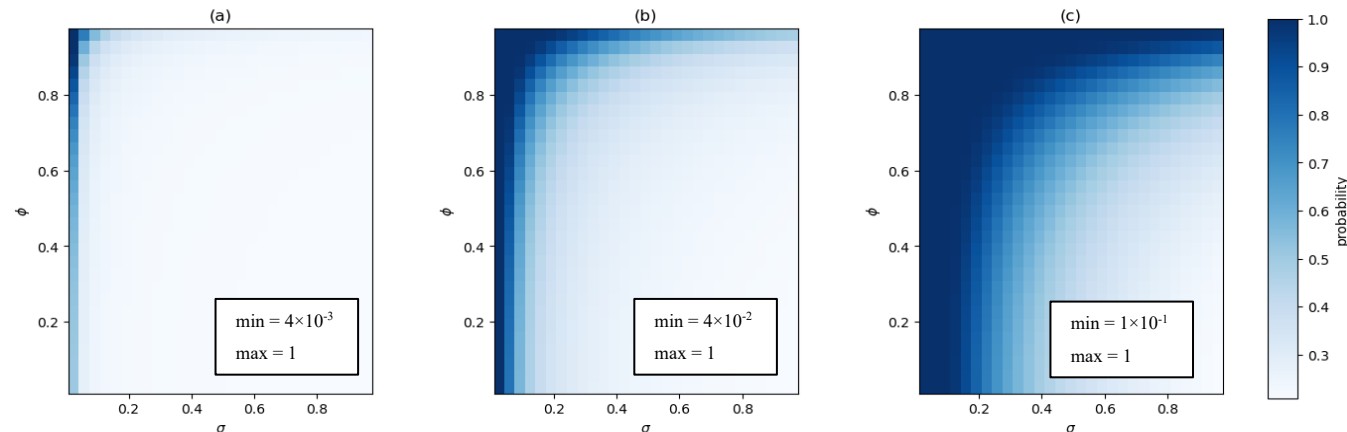

**Figure 2. Conditional probabilities to find a measured value $x_t$ given $x_{t-1}$ for three different numerical resolutions, i.e. (a) *res* = 0.01, (b) *res* = 0.1 and (c) *res* = 0.5. In this figure, the $\sigma$ and $\varnothing$ are ranged from 0.01 to 0.99.**

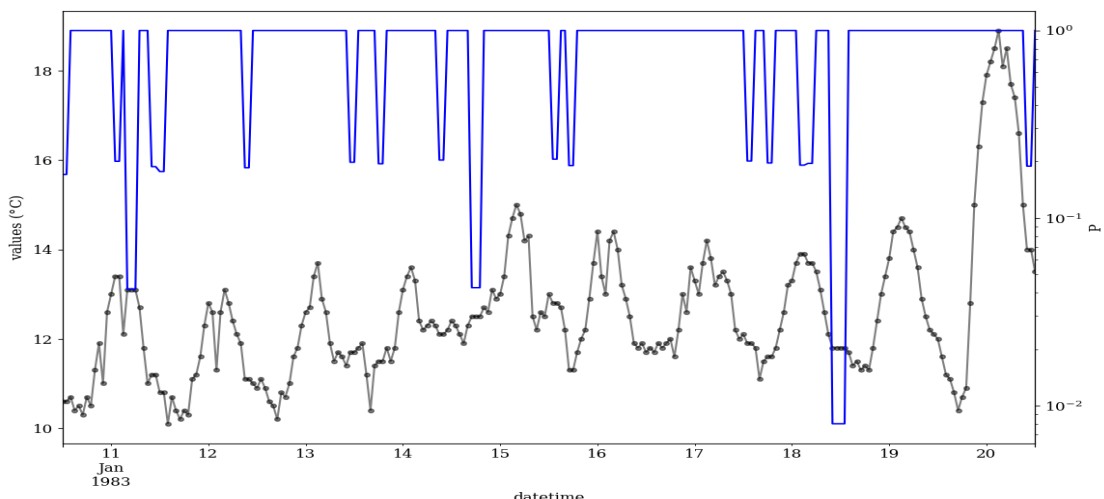


**Figure 3. Temperature time series at the Cape Grim station (40°68′ S, 144°69′ E) from 10$^{th}$ to 20$^{th}$ of January 1983. Black and blue lines show the temperature value (°C) and its associated probability, *P* in Eq. (9), respectively. In this figure, the time is shown in UTC. The data were retrieved from the TOAR database.**





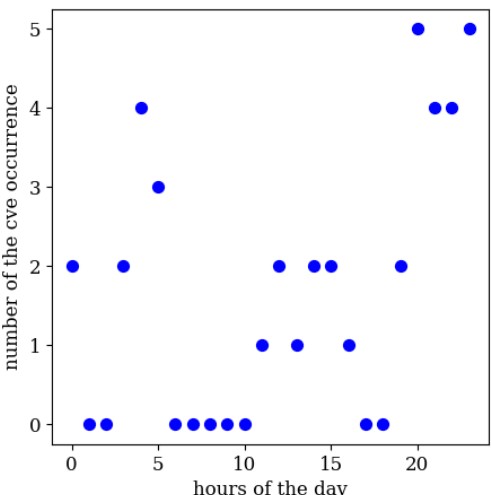

**Figure 4. The number of the CVEs occurring for the different hours in a day, i.e., $h$ = {0...23}, for the temperature time series shown in Fig. 3.**

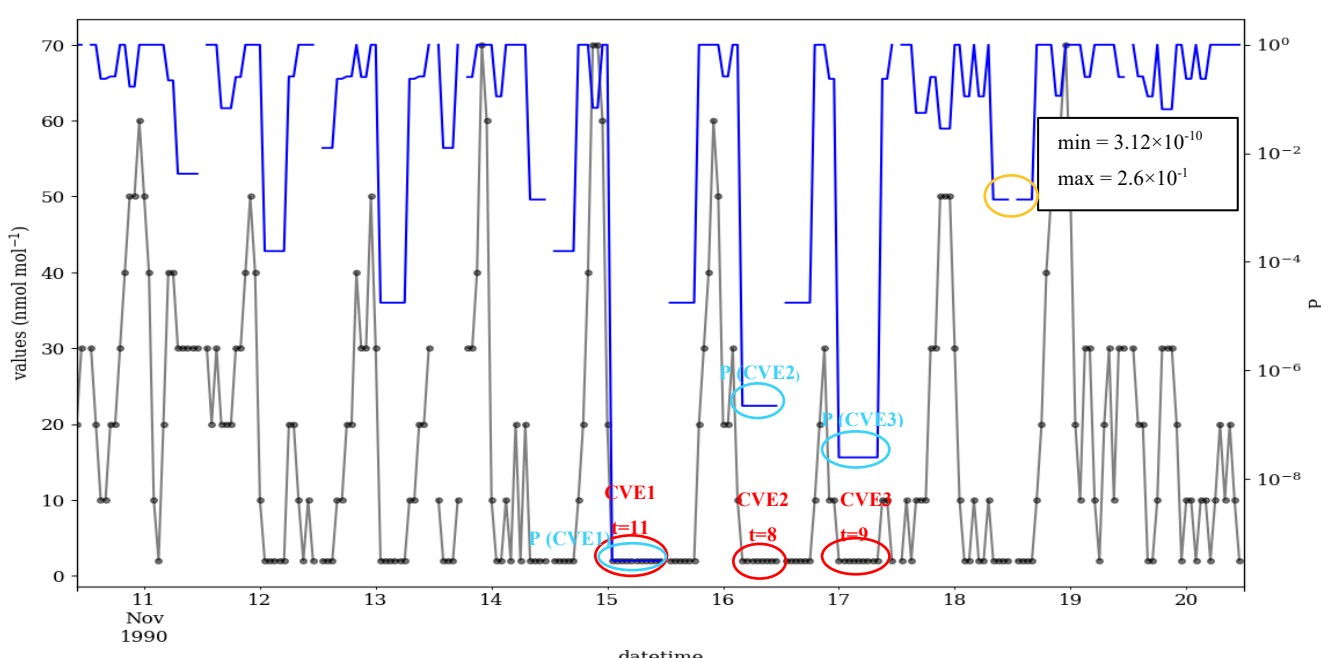

**Figure 5. Time series of ozone mixing ratio at the Azusa station, California, from 10$^{th}$ to 20$^{th}$ November, 1990 (black) and CVT test results (blue). During this period, the data were recorded in intervals of 8 ppb, i.e., *res* = 8, so that valid CVEs are frequent. In total, this time series contains 45 CVEs as 27, 6, 3, 3, 1, 1, 1, and 1 episode with the $t$ = 2, 3, 4, 5, 6, 8, 9, and 11, respectively. The red circles (or ovals) highlight three examples of zero-ozone incidents (here 2 ppb) with a large length ($t \geq 8$) in this series. The cyan circles highlight the probability of the respective CVEs. The orange circle highlights a CVEs with a length of 4 that contain a gap of missing data points.**





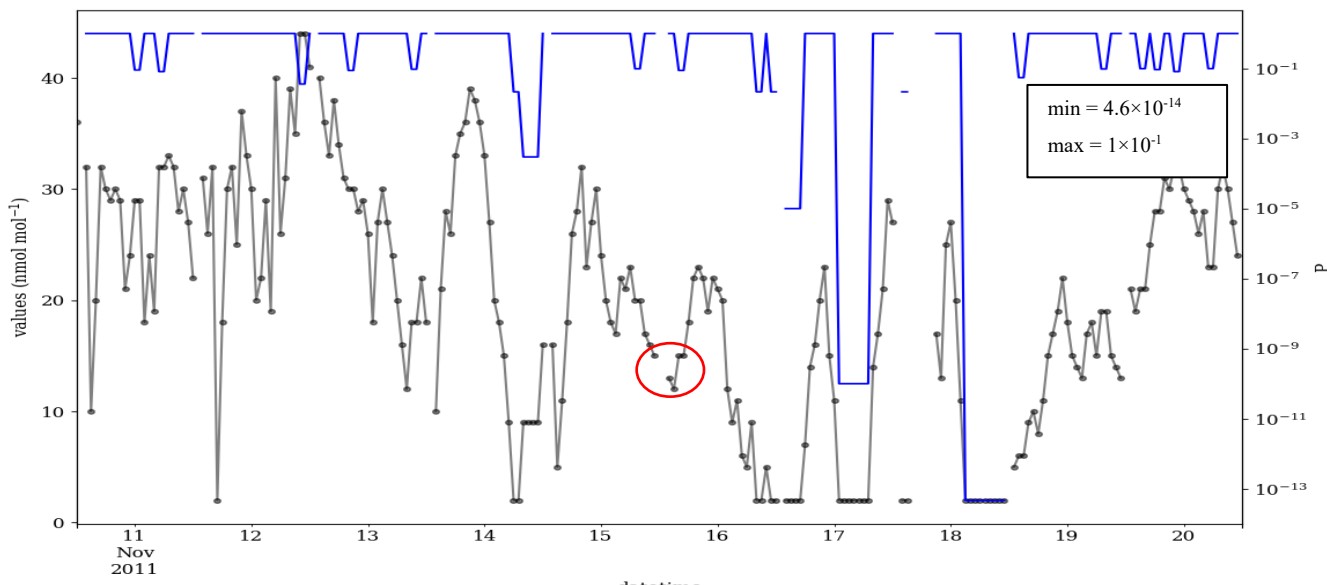

**Figure 6. As Fig. 5, but from 10th to 20th November 2011, when the data were recorded with a numerical resolution of 1 ppb, i.e., res = 1. The red circle shows one example of missing data points in the data time series. The μ, σ, and Ø of the data in this figure are 19.9, 10.73 and 0.84, respectively.**

**Appendix (A)**

The inference of conditional probability of bivariate normal distribution

$$\frac{f(x_{k-1}, x_k)}{f(x_{k-1})} =$$

$$\frac{\frac{1}{2\pi\sigma^2\sqrt{1-\emptyset^2}} \exp\left(-\frac{1}{2(1-\emptyset^2)}\left[\frac{(x_{k-1}-\mu)^2}{\sigma^2} + \frac{(x_k-\mu)^2}{\sigma^2} - \frac{2\emptyset(x_{k-1}-\mu)(x_k-\mu)}{\sigma^2}\right]\right)}{\frac{1}{\sigma\sqrt{2\pi}} \exp\left(-\frac{1}{2}\left[\frac{(x_{k-1}-\mu)^2}{\sigma^2}\right]\right)} =$$

$$\frac{1}{\sigma\sqrt{2\pi(1-\emptyset^2)}} \exp\left(-\frac{1}{2(1-\emptyset^2)}\left[\frac{(x_{k-1}-\mu)^2}{\sigma^2} + \frac{(x_k-\mu)^2}{\sigma^2} - \frac{2\emptyset(x_{k-1}-\mu)(x_k-\mu)}{\sigma^2} - \frac{(1-\emptyset^2)(x_{k-1}-\mu)^2}{\sigma^2}\right]\right) =$$

$$\frac{1}{\sigma\sqrt{2\pi(1-\emptyset^2)}} \exp\left(-\frac{1}{2(1-\emptyset^2)}\left[\frac{\emptyset^2(x_{k-1}-\mu)^2}{\sigma^2} + \frac{(x_k-\mu)^2}{\sigma^2} - \frac{2\emptyset(x_{k-1}-\mu)(x_k-\mu)}{\sigma^2}\right]\right)$$

$\sim N(\mu + \emptyset(c - \mu), (1-\emptyset^2)\sigma^2)$, given $x_{k-1} = c$.



## Appendix (B)

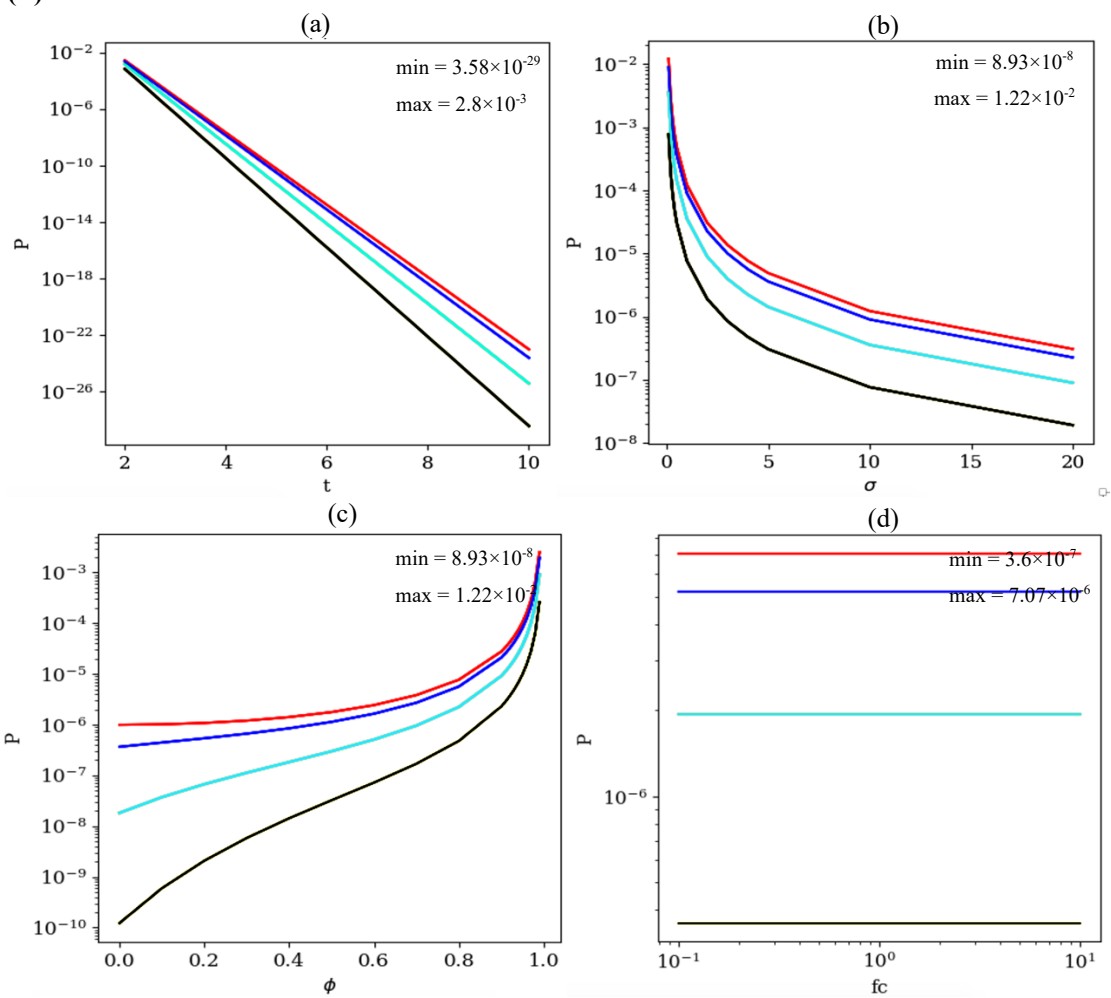

**Figure B1.** Sensitivity of $P$ to the (a) CVEs' length, i.e., $t$ = 2, 3, 4, 5, 6, 7, 8, 9, and 10. Other parameters are fixed as $\mu$ = 10, $\sigma$ = 4, $\emptyset$ = 0.8, and c-$\mu$ = 0, 4, 8, and 12. (b) standard deviation, i.e., $\sigma$ = 0.1, 0.2, 0.3, 0.4, 0.5, 1, 2, 3, 4, 5, 10, and 20. Other parameters are fixed as $\mu$ = 10, $t$ = 3, $\emptyset$ = 0.8, and c-$\mu$ = 0, 4, 8, and 12. (c) lag-1 autocorrelation, i.e., $\emptyset$ = 0., 0.1, 0.2, 0.3, 0.4, 0.5, 0.6, 0.7, 0.8, 0.9, 0.91, 0.92, 0.93, 0.94, 0.5, 0.96, 0.97, 0.98, and 0.99. Other parameters are fixed as $\mu$ = 10, $\sigma$ = 4, $t$ = 3, and c-$\mu$ = 0, 4, 8, and 12. (d) Sensitivity of $P$ to scaling factor, i.e., $fc$ = 0.1, 0.2, 0.5, 1, 2, 5, and 10. Other parameters are fixed as $\emptyset$ = 0.8 and $t$ = 3. The same colour codes are applied as that in Fig. 1.





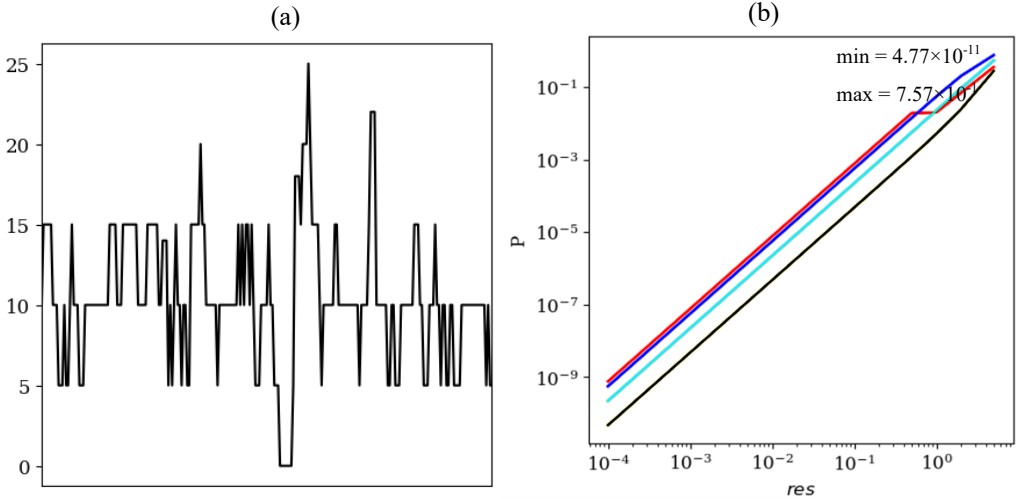

**Figure B2. (a) The modified time series (*res* = 5) where *ref* time series were resampled with rounding to the nearest of 5. That includes more CVEs than the *ref* in Fig. 1. (b) Sensitivity of *P* to the digital numerical precision, i.e., *res* = 0.0001, 0.0002, 0.0005, 0.001, 0.002, 0.005, 0.01, 0.02, 0.05, 0.1, 0.2, 0.5, 1, 2, and 5. Other parameters are fixed as μ = 10, σ = 4, ∅ = 0.8, *t* = 3, and c-μ = 0, 4, 8, and 12. The same colour codes are applied as that in Fig. 1.**

**Appendix (C)**

If the data are normalized, i.e., $(x - x_{min}) / (x_{max} - x_{min})$:

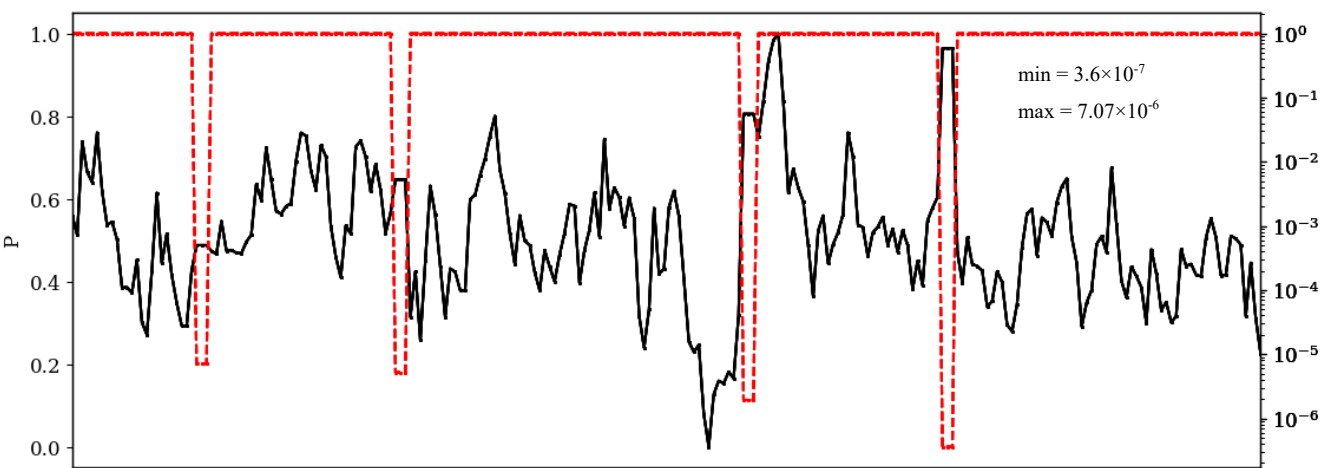

**Figure C1. As Fig 1. But the data time series are normalized, $\mu = 0.5, \sigma = 0.15, \varnothing = 0.8,$ and $res = 0.004$**





If the data are standardized, i.e. (x-μ) / σ:

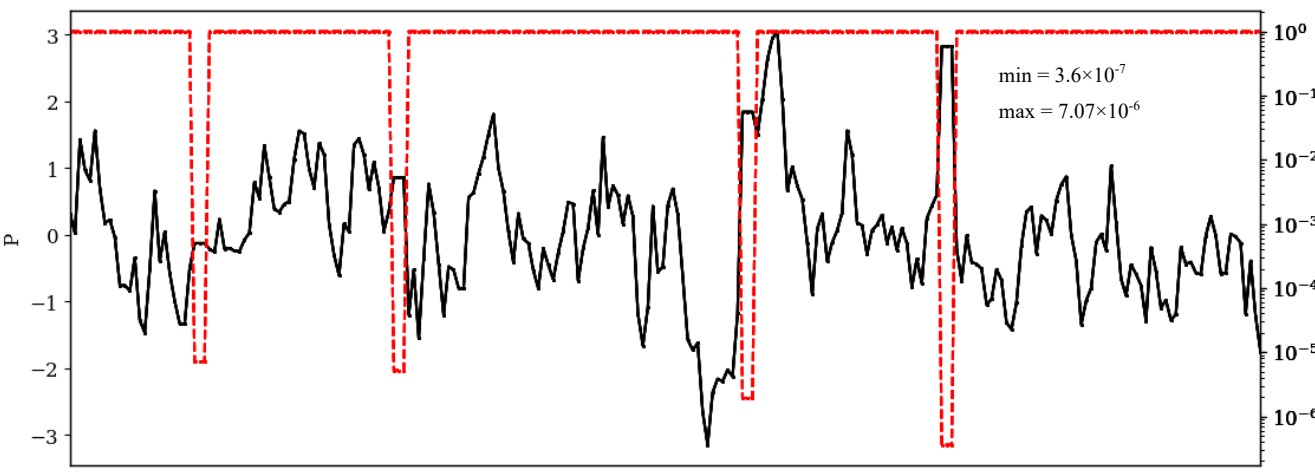

**Figure C2. As Fig 1. But the data time series are standardized, $\mu = -0.07, \sigma = 0.94, \emptyset = 0.8$, and $res = 0.002$.**

**Appendix (D)**

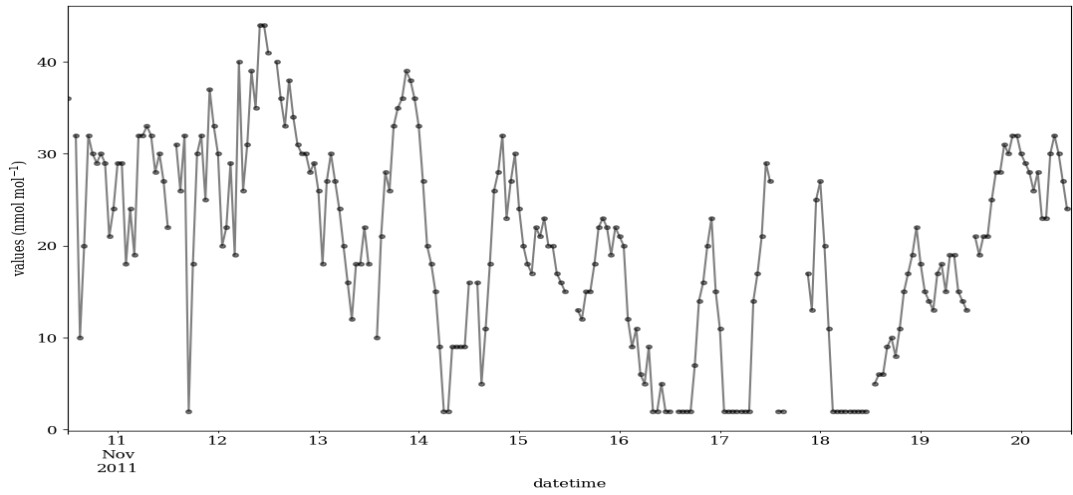

**Figure D1. Time series of ozone mixing ratio at the Azusa station, California, from 10th to 20th November, 2011. During this period, the data were recorded in intervals of 1 ppb, i.e., $res = 1$. $\mu = 19.9, \sigma = 10.73, and \emptyset = 0.84$.**



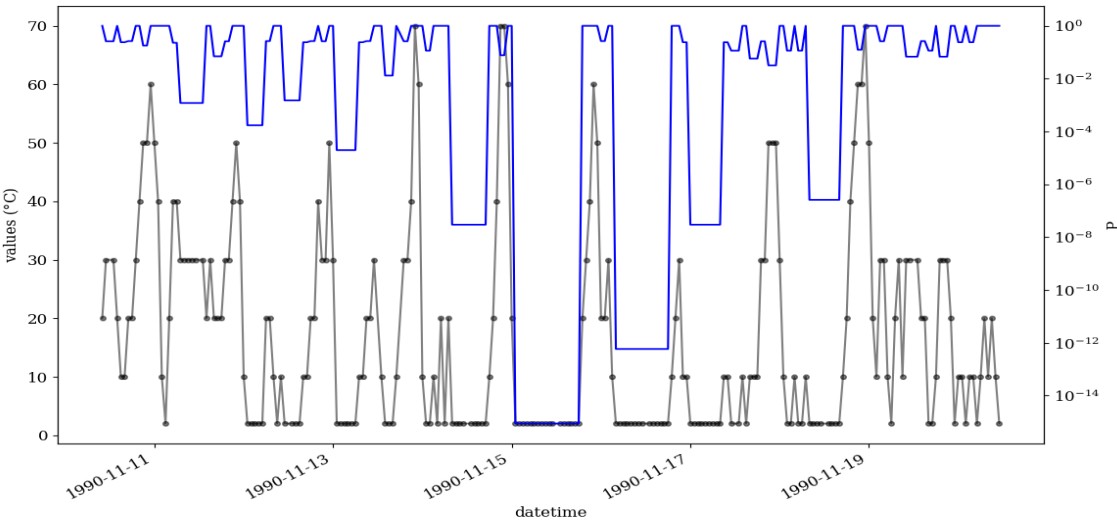

**Figure D2. As Fig. 6, but the missing values are not treated. So, the orange circle shows two CVEs, which have been merged to one**
**incident with a longer length ($t = 8$).**

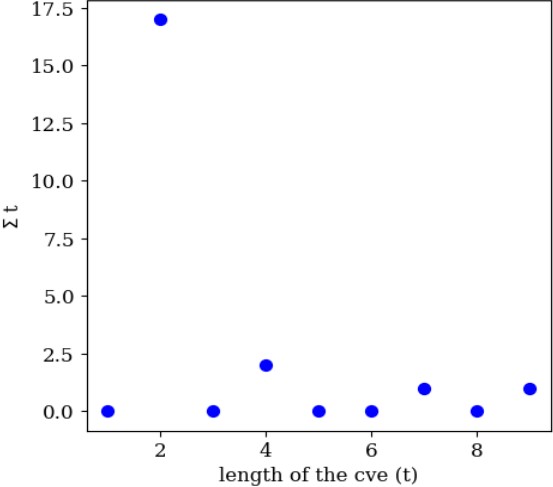

**Figure D3. Number of CVEs ($\sum t$) of the different length, i.e., $t = \{0, \ldots, 9\}$, for the ozone time series of year 2011 (shown in Fig. 6).**


