# Peer review of "A data-driven persistence test for robust (probabilistic) quality control of measured environmental time series: constant value episodes"

_Atmospheric Measurement Techniques, 2022_

## Author Response (AR1)

**Authors' response to reviewer #1**

Please note that in this document, colors' codes are the referee's comments in black, the author's responses in blue. The author's changes in the manuscripts are shown in blue bold. Line numbers and figures are referred based on their numbers in the revised manuscript.

The author highly acknowledges the referee for spending time and providing punctual comments.

- Page 2, line 46: please double check the reference provided for AutoQC – is there are year of publication? Is a link to a github site sufficient reference, or should the reference to be to an actual document?

  That is right. A reference has been available recently. It was added as:

  L46: **"AutoQC (Good et al., 2022)"**

  L413-L414: **"Good, S., Mills, B. and Castelao, G.: AutoQC: Automatic quality control analysis for the international quality controlled ocean database, Zenodo [code], https://doi.org/10.5281/zenodo.5832003, 2022."**

- Page 2, line 47: add "and" before "WMO-AWS"

  It was added as:

  L47: **"and WMO-AWS …"**

- Page 2, lines 57 – 58: suggested rewording as "... in a specific 35-year long ozone time series, the occurrence ..."

  It was changed as:

  L57-L58: **"in a specific 35-year long ozone time series, the …"**

- Page 3, line 70: suggest changing "being valid data" to "representing valid data"

  It was changed as:

  L73-L74: **"a CVE representing valid data …"**

- Page 3, line 80: suggest changing "the method" to "the proposed method"

  It was changed as:

  L84: **"Before describing the proposed method, …"**

- Page 3, line 91: suggest changing "... significant digits that an observation is recorded" to "... significant digits with which an observation is recorded"

  It was changed as:

  L95: **"which is the number of significant digits with which an observation is recorded ..."**

- Page 3, lines 91 – 94: this example mentions an O3 reporting precision of 10 ppb, but the later Azusa ozone example example in Section 4.2 is for a reporting precision of 8 ppb. It might make sense to align the introduction of this issue in this section with the example provided in the later section. In my recollection, another pollutant in the EPA AQS database for which reporting precision has changed over time since the 1980s is CO so this could be mentioned here as well and an additional example could be added to Section 4, but of course you would want to double check this first, my recollection could be incorrect.

  That is right and thank you for the suggestions. They were applied as:

  L95-L98: **"For example, historical measurements of ground-level ozone (Azusa station) in the EPA Air Quality System (AQS) in the 1980s were reported with a resolution of 8 parts per billion (ppb). Another pollutant in the EPA AQS database for which reporting precision has changed over time since 1980 is carbon monoxide at the Fresno station (California state). So,"**

  L248-L251: **"Data series of carbon monoxide at the Fresno station (36.78° N, 119.77° W) were obtained from the EPA AQS database. This data was reported with a precision of 1 ppm in 1980 and later changed to a higher precision of 0.001 ppm, depending on the measurements' method (e.g., instrumental-nondispersive infrared, instrumental-gas filter correlation Teledyne API 300 EU)."**

  L312-L320: "**4.3 Carbon Monoxide**
  **Exposure to elevated carbon monoxide harms the human body, in particular, those who suffer from heart diseases. This air pollutant also affects some greenhouse gases, e.g., carbon dioxide and ozone, which are linked to climate change and global warming. A 10-day example of the measured carbon monoxide at the Fresno station is shown in Fig. 7. Despite of high precision of the data for the year 2022 (res = 0.001, see Fig. D4), data were recorded with a resolution of 1 ppm in 1980. This data contains fewer CVEs but with a larger t (19 CVEs with t = 2 ... 34) in comparison to the ozone series in Fig. 5. That could associate with a longer lifetime of carbon monoxide than that of ozone. This reflects that most of the CVEs in the carbon monoxide series are valid. The CVT discerns this and estimates a larger P for this data, in which the smallest P is 0.001 for the CVEs with t= 14 and values of 0 ppm."**

  L328-L329: "**such as air temperature, ozone mixing ratio, or carbon monoxide as time series with serial dependence."**

  L363: **"U.S. EPA AQS for providing the ozone time series at Azusa and carbon monoxide data at Fresno are appreciated."**

L590: **"Figure 7. Time series of carbon monoxide at the Fresno station, California, from 1$^{st}$ to 11$^{th}$ January 1980 (black) and CVT test results (blue). During this period, the data were recorded in intervals of 1 ppm, i.e., res= 1, so that valid CVEs are frequent. In total, this time series contains 19 CVEs as 1, 1, 1, 1, 2, 2, 1, 2, 1, 1, 1, 3, and 2 episodes with the t= 34, 27, 21, 18, 15, 14, 12, 11, 10, 5, 4, 3, and 2, respectively. The μ, σ, and  of the data in this figure are 0.79, 0.45, and 0.65, respectively."**

L640: **"Figure D4. As Fig. 7, but from 1$^{st}$ to 11$^{th}$ January 2022, when the data were recorded with a numerical resolution of 0.001 ppm, i.e.,  res= 0.001. This series shows three CVEs with a length of 2, i.e., t = 2. The μ, σ, and  of the data in this figure are 0.62, 0.4, and 0.88, respectively."**

- Page 8, line 228: please change "from the 1980" to "from 1980" or "from the 1980s", whichever is appropriate.

  It was changed as:

  L244: **"from the 1980s, …"**

- Page 8, lines 228 – 299: could you please elaborate on the statement "data were recorded with a resolution of 8 ppb, depending on environmental conditions"? How exactly did environmental conditions impact recording resolution? Does this statement imply that recording resolution fluctuated over time depending on conditions, with sometimes higher and sometimes lower resolution, or was there a specific point in time (or several points in time) when resolution increased from 8 ppb to 1 ppb? If resolution fluctuated repeatedly over time, can this be handled properly by the proposed method if the analysis period includes several periods of fluctuation resolution?

  Great question. Environmental conditions refer to the periods, in which the instruments were not so developed and could not measure the values with high precisions. Resolution is changed over time and mostly depends on the instruments' techniques. They are supposed to be advanced in time and measure the data with higher resolutions. One exception to that is the measured CO at Jefferson stations (Alabama state), in which the resolutions have been changed over time as:
  in 1980 (res=0.1): INSTRUMENTAL - NONDISPERSIVE INFRARED
  in 2017 (res=0.001): INSTRUMENTAL - Gas Filter Correlation Teledyne API 300 EU
  in 2022 (res=0.1): INSTRUMENTAL - GAS FILTER CORRELATION CO ANALYZER
  The method can handle any resolution, but the current version of the code only resolves the highest resolution of the data series.

- Page 8, line 243: suggest changing "when the CVEs were to be flagged" to "if the CVEs were to be flagged". Also, out of curiosity, were any of these events (manually) flagged as suspicious by the QC applied by the organization providing this data? If not, the argument that one would err in 1 out of 5 cases by flagging these events is purely hypothetical.
  It was changed as:

  L261: **"which means that, if the CVEs were …"**

Good question. Not actually, we did not receive the QC report of the data. The mentioned argument is hypothetical.

- Page 9, line 256: change "10-days examples" to "10-day example"

It was changed as:

L285: **"a 10-day example …"**

- Page 9, line 265: fix typo "filtered as as"

It was not found. It was written as:

L294: "**filtered as an …**"

- Page 9, line 277: suggest changing the beginning of the sentence to "Environmental time series are valuable and essential data sources …"

It was changed as:

L322: **"Environmental time series are valuable and essential data sources for …"**

Please note that in this document, colors' codes are the referee's comments in black, the author's responses in blue. The author's changes in the manuscripts are shown in blue bold. Line numbers and figures are referred based on their numbers in the revised manuscript.

The author highly acknowledges the referee for spending time and providing punctual comments.

- How does the presented method relate to data assimilation methods in numerical weather prediction? (e.g. ERA5) Here, measurement data is regularly assimilated in the model run based on measurement errors derived for each measurement station. CVEs should be in an issue there as well, no?

  In assimilation processes, the measured (observed) data are used after checking their qualities, which are covered via various tests such as outliers, range, etc. CVEs are often flagged as erroneous data and removed from the series, while they can reflect natural environmental phenomena, or be caused by low recording precision of instruments (back in 1980). That can be an issue, especially for historical reanalysis products, which are more based on in situ data (fewer satellite data are available). So, if the CVEs exclude from the data, the results of the (re)analysis can become biased.

  This point was added as:

  L69-L70: "**That can be an issue in (re)analysis products (Inness et al., 2019; Hersbach et al., 2020), where assimilation processes reduce misfits between observations and their modeled values.**"

  L421-L423: "**Hersbach, H., Bell, B., Berrisford, P., Hirahara, S., Horányi, A., Nicolas, J., Peubey, C., Radu, R., Bonavita, M., Dee, D., Dragani, R., Flemming, J., Forbes, R., Geer, A., Hogan, R. J., Janisková, H. M., Keeley, S., Laloyaux, P., Cristina, P. L., and Thépaut, J.: The ERA5 global reanalysis, 1999–2049, https://doi.org/10.1002/qj.3803, 2020.**"

  L438-L441: "**Inness, A., Ades, M., Agustí-Panareda, A., Barr, J., Benedictow, A., Blechschmidt, A. M., Jose Dominguez, J., Engelen, R., Eskes, H., Flemming, J., Huijnen, V., Jones, L., Kipling, Z., Massart, S., Parrington, M., Peuch, V. H., Razinger, M., Remy, 440 S., Schulz, M., and Suttie, M.: The CAMS reanalysis of atmospheric composition, Atmos. Chem. Phys., 19, 3515–3556, https://doi.org/10.5194/acp-19-3515-2019, 2019.**"

- If I understood correctly, sigma is a key quantity with a strong influence on the probability of CVEs. Doesn't this create a dependency on climate regions? E.g. in polar regions, the diurnal cycle of the temperature in summer could be quite high, but coastal regions in that area with a dense fog might have morning periods when the temperature is rather constant. Would this combination raise false flags?

  Sigma is one of the effecting parameters ($\sigma$, $\varphi$, $\mu$, etc.) on the probability ($P$). If the diurnal cycle of temperature in polar contains a larger $\sigma$ than that in coastal regions, the $P$ will be less in polar than in coastal sites, assuming all other parameters are constant (as shown in Fig. B1. b). So, in coastal stations, where

the CVEs occur more often, the *P* is larger, to avoid those data from being flagged. Nevertheless, the flags are defined based on *P*, one may use a lower threshold for *P* in polar regions than the coastal sites in the region.

The text is modified as:

L273-L277: "**Other criteria for selecting a threshold for *P* could be climate regions. In polar regions, the diurnal cycle of the temperature in summer could be quite high, but coastal sites in that area with a dense fog might have morning periods when the temperature is rather constant. The first shows a larger $\sigma$ than the latter, so the *P* will be less in the polar than the coastal sites, assuming all other parameters are constant (as shown in Fig. B1. b). One may adopt a smaller threshold for *P* in polar than coastal sites.**"

- Same holds for the "distance to the mean" – for the same climatological region, constant temperature values at night or at the day when the diurnal cycle reaches maximum or minimum the CVT would give CVEs a lower probability as they are further from the mean. However, CVEs are more likely when the diurnal cycle reaches an extreme value, i.e. turning point.

  That is right. The larger distance from the mean ($\mu - c$) gives a lower *P*, assuming all other parameters are constant. So, the *P* of CVEs at extremums can be less than the CVEs with the same *t*.

  This point was mentioned as:

  L277-L279: "**Or for the same climatological region, constant temperature values at night or during the day, when the diurnal cycle reaches maximum or minimum, the CVT would give CVEs a lower probability as they are further from the mean (larger c-$\mu$). So, the *P* of the CVEs at extremums can be less than the CVEs with the same *t* in this series.**"

- Would the Probability for a CVE change, if the analysis of the temperature time series in Figure 3 is done in K instead of °C?

  No, it will not be affected as the *P* depends on the parameters such as $\sigma$, $\varphi$, $\mu - c$, *t, and res,* which are not varied by °C to °K conversion.

  The caption of Fig. 3 was modified as:

  L571: "**The *P* is not affected by the unit conversion, i.e., °C to °K.**".

- [line 263 to 265]: "The CVT can recognize such cases and the associated probabilities are $3.12 \times 10^{-10}$, $2.22 \times 10^{-7}$, and $2.48 \times 10^{-8}$, for the CVE1, CVE2 and CVE3, respectively. That would prevent such (valid) values from being flagged or filtered as an erroneous data."

If I understand right, a low probability of a CVE would lead to a flagging of those values and thus the flagging of valid values would not be prevented – Maybe the author could clarify this section a little further. Where would the author suggest the P threshold for the ozone time series?

There is no suggestion for the threshold of *P*. That is mentioned based on the comparison of two parts of the time series, i.e., the year 1990 and 2011, where the minimum *P* for the CVEs are $3.12 \times 10^{-10}$ and $4.6 \times 10^{-14}$, respectively. The CVEs in the second part of the time series (i.e., 2011) are rare. This sparse incident arises the suspicion of data analysts and could be used as a threshold of *P* (e.g., $10^{-14}$) for this series.

The section was modified as:

L294-L295: "**as an erroneous data, in contrast to the second part of the time series in Fig. 6 (for the year 2011), which exhibits …**"

A typo was fixed as:

L298: "**The estimated *P* for that incident is $4.6 \times 10^{-14}$.**"

- [line 228 and line 232] "the 1980" – either just "1980" or "the 1980s"

  It was changed as:

  L247: "**… an extensive record back into 1980.**"

- [line 277] "Environmental time series is..." - "Environmental time series are..."

  It was changed as:

  L322: "**Environmental time series are valuable …**"

---

## Editor Decision (ED1)

[revised manuscript text omitted]
{\dfrac{1}{2\pi\sigma^2\sqrt{1-\emptyset^2}} \exp\left(-\dfrac{1}{2(1-\emptyset^2)}\left[\dfrac{(x_{k-1}-\mu)^2}{\sigma^2} + \dfrac{(x_k-\mu)^2}{\sigma^2} - \dfrac{2\emptyset(x_{k-1}-\mu)(x_k-\mu)}{\sigma^2}\right]\right)}{\dfrac{1}{\sigma\sqrt{2\pi}} \exp\left(-\dfrac{1}{2}\left[\dfrac{(x_{k-1}-\mu)^2}{\sigma^2}\right]\right)} =$$

$$\frac{1}{\sigma\sqrt{2\pi(1-\emptyset^2)}} \exp\left(-\frac{1}{2(1-\emptyset^2)}\left[\frac{(x_{k-1}-\mu)^2}{\sigma^2} + \frac{(x_k-\mu)^2}{\sigma^2} - \frac{2\emptyset(x_{k-1}-\mu)(x_k-\mu)}{\sigma^2} - \frac{(1-\emptyset^2)(x_{k-1}-\mu)^2}{\sigma^2}\right]\right) =$$

600    $$\frac{1}{\sigma\sqrt{2\pi(1-\emptyset^2)}} \exp\left(-\frac{1}{2(1-\emptyset^2)}\left[\frac{\emptyset^2(x_{k-1}-\mu)^2}{\sigma^2} + \frac{(x_k-\mu)^2}{\sigma^2} - \frac{2\emptyset(x_{k-1}-\mu)(x_k-\mu)}{\sigma^2}\right]\right)$$

$$\sim N(\mu + \emptyset_{(c-\mu)}, (1-\emptyset^2)\sigma^2), \text{ given } x_{k-1} = c.$$

605

610

[Figure]

Figure B1. Sensitivity of $P$ to the (a) CVEs' length, i.e., $t$ = 2, 3, 4, 5, 6, 7, 8, 9, and 10. Other parameters are fixed as $\mu$ = 10, $\sigma$ = 4, $\emptyset$ = 0.8, and c-$\mu$ = 0, 4, 8, and 12. (b) standard deviation, i.e., $\sigma$ = 0.1, 0.2, 0.3, 0.4, 0.5, 1, 2, 3, 4, 5, 10, and 20. Other parameters are fixed as $\mu$ = 10, $t$ = 3, $\emptyset$ = 0.8, and c-$\mu$ = 0, 4, 8, and 12. (c) lag-1 autocorrelation, i.e., $\emptyset$ = 0., 0.1, 0.2, 0.3, 0.4, 0.5, 0.6, 0.7, 0.8, 0.9, 0.91, 0.92, 0.93, 0.94, 0.5, 0.96, 0.97, 0.98, and 0.99. Other parameters are fixed as $\mu$ = 10, $\sigma$ = 4, $t$ = 3, and c-$\mu$ = 0, 4, 8, and 12. (d) Sensitivity of $P$ to scaling factor, i.e., $fc$ = 0.1, 0.2, 0.5, 1, 2, 5, and 10. Other parameters are fixed as $\emptyset$ = 0.8 and $t$ = 3. The same colour codes are applied as that in Fig. 1.

**Figure B2. (a) The modified time series (*res* = 5) where *ref* time series were resampled with rounding to the nearest of 5. That includes more CVEs than the *ref* in Fig. 1. (b) Sensitivity of *P* to the digital numerical precision, i.e., *res* = 0.0001, 0.0002, 0.0005, 0.001, 0.002, 0.005, 0.01, 0.02, 0.05, 0.1, 0.2, 0.5, 1, 2, and 5. Other parameters are fixed as μ = 10, $\sigma$ = 4, Ø = 0.8, *t* = 3, and c-μ = 0, 4, 8, and 12. The same colour codes are applied as that in Fig. 1.**

**Appendix (C)**

If the data are normalized, i.e., $(x - x_{min}) / (x_{max} - x_{min})$:

[Figure]

**Figure C1. As Fig 1. But the data time series are normalized, $\mu = 0.5, \sigma = 0.15, \emptyset = 0.8,$ and $res = 0.004$**

625     If the data are standardized, i.e. (x-μ) / σ:

[Figure]

$min = 3.6\times10^{-7}$
$max = 7.07\times10^{-6}$

**Figure C2. As Fig 1. But the data time series are standardized, $\mu = -0.07$, $\sigma = 0.94$, $\emptyset = 0.8$, and $res = 0.002$.**

**Appendix (D)**

[Figure]

630     **Figure D1. Time series of ozone mixing ratio at the Azusa station, California, from 10[th] to 20[th] November, 2011. During this period, the data were recorded in intervals of 1 ppb, i.e., $res = 1$. $\mu = 19.9$, $\sigma = 10.73$, $and$ $\emptyset = 0.84$.**

[Figure]

**Figure D2. As Fig. 6, but the missing values are not treated. So, the orange circle shows two CVEs, which have been merged to one**
635     **incident with a longer length ($t = 8$).**

[Figure]

**Figure D3. Number of CVEs ($\sum$ t) of the different length, i.e., $t = \{0, …, 9\}$, for the ozone time series of year 2011 (shown in Fig. 6).**

[Figure]

**Figure D4. As Fig. 7, but from 1ˢᵗ to 11ᵗʰ January 2022, when the data were recorded with a numerical resolution of 0.001 ppm, i.e., *res* = 0.001. This series shows three CVEs with the length of 2, i.e., *t* = 2. The μ, σ, and ∅ of the data in this figure are 0.62, 0.4, and 0.88, respectively.**